# Parallel Frequency Function-Deep Neural Network for Efficient Approximation of Complex Broadband Signals

**DOI:** 10.3390/s22197347

**Published:** 2022-09-28

**Authors:** Zhi Zeng, Pengpeng Shi, Fulei Ma, Peihan Qi

**Affiliations:** 1School of Mechano-Electronics Engineering, Xidian University, Xi’an 710071, China; 2School of Civil Engineering & Institute of Mechanics and Technology, Xi’an University of Architecture and Technology, Xi’an 710055, China; 3State Key Laboratory for Strength and Vibration of Mechanical Structures, Shaanxi Engineering Research Center of NDT and Structural Integrity Evaluation, School of Aerospace, Xi’an Jiaotong University, Xi’an 710049, China; 4School of Telecommunications Engineering, Xidian University, Xi’an 710071, China

**Keywords:** PFF-DNN, spectral bias, broadband signals, fast Fourier analysis

## Abstract

In recent years, with the growing popularity of complex signal approximation via deep neural networks, people have begun to pay close attention to the spectral bias of neural networks—a problem that occurs when a neural network is used to fit broadband signals. An important direction taken to overcome this problem is the use of frequency selection-based fitting techniques, of which the representative work is called the PhaseDNN method, whose core idea is the use of bandpass filters to extract frequency bands with high energy concentration and fit them by different neural networks. Despite the method’s high accuracy, we found in a large number of experiments that the method is less efficient for fitting broadband signals with smooth spectrums. In order to substantially improve its efficiency, a novel candidate—the parallel frequency function-deep neural network (PFF-DNN)—is proposed by utilizing frequency domain analysis of broadband signals and the spectral bias nature of neural networks. A substantial improvement in efficiency was observed in the extensive numerical experiments. Thus, the PFF-DNN method is expected to become an alternative solution for broadband signal fitting.

## 1. Introduction

An artificial neural network, also known as a neural network, refers to a mathematical model that imitates animal neural network behavior [1], which is essentially a high-dimensional complex mapping model wherein adjusting network weights allows for feature fitting. A neural network’s basic building blocks are a neuron model, a parameterized model nested by a scalar linear function, and a monotonic nonlinear function (activation function), where the coefficients in the linear function are the connection weights between neurons. Neurons connecting according to a specific topology form a neural network. One of the primary networks is the single-layer network composed of multiple neurons in parallel. Multiple single-layer networks can be stacked to obtain a multi-layer network and further expanded into a deep neural network that contains multiple multi-layer networks. Some advanced neural network models are designed to meet the needs of engineering practice [2,3,4,5]. For example, convolutional neural networks (CNN) and their evolution models have achieved unprecedented success in computer vision due to their powerful feature extraction capabilities and are widely used in security monitoring, autonomous driving, human-computer interaction, augmented reality, and many other fields. Additionally, recurrent neural networks, especially the long-short-term memory model (LSTM), have become mainstream tools in the fields of automatic translation, text generation, and video generation in just a few years.

The related research of the universal approximation [6,7,8,9,10] indicates that assuming sufficient neurons and suitable weights, the neural network could approximate any continuous signals on a compact subset of ℝ^n^ with arbitrary precision. However, it is not easy to obtain these appropriate weights via training for a complex network with too many neurons. The convergence speed of the neural network is related to the frequency spectrum of the fitted signal [11]. As shown in Figure 1, the neural network first learns the low-frequency components during the training process. The relationship between the convergence speed and the frequency of the fitted signal has been quantitatively analyzed [12]. When a network is applied to fit a signal, the required training time increases exponentially as the component’s central frequency increases. The spectral bias of the convergence speed in network training leads to unbearable training times for fitting the high-frequency components in broadband signals. In recent years, differential equation solving based on physical knowledge constraints has been successful in many applications, but multi-scale and multi-physics problems need further development. One of the core problems is that it is difficult for fully connected neural networks to learn high-frequency functions, that is, the spectral bias problem [13]. Solving this problem has become one of the core bottlenecks for the further development of A.I. technology.

To this end, Cai et al. proposed the PhaseDNN method for fitting complex signals with high-frequency components via the combination of parallel frequency band extraction and frequency shifting techniques [14]. Numerous numerical experimental results have indicated that the PhaseDNN method successfully avoids the computational cost disaster when fitting signals with high-frequency components. However, although the method has good approximation efficiency for broadband signals, we found in our numerical experiments that for a large number of typical signals, especially those with smooth frequency domain distribution, the operation of fitting the inverse transformed signals can be further optimized, thus greatly improving the neural network fitting efficiency.

Therefore, our paper is dedicated to investigating a more efficient candidate method for fitting complex signals—signals with smooth frequency domain distribution. To reach this goal, a parallel frequency function-deep neural network (PFF-DNN) is proposed by utilizing the fast Fourier analysis of broadband signals and the spectral bias nature of neural networks. The effectiveness and efficiency of the proposed method are verified based on detailed experiments for six typical broadband signals. The discussion shows how the neural network adaptively constructs low-frequency smooth curves to interpolate discrete signals. This adaptive low-frequency approximation makes it possible to fit discrete frequency domain signals accurately.

The paper is organized as follows. Related works and the proposed method are introduced in Section 2. Extensive numerical experiments are presented in Section 3. Finally, the paper is concluded in Section 4.

## 2. Methods

### 2.1. Recent PhaseDNN

The research mentioned above indicates that it is difficult for neural networks to directly and accurately fit signals with high-frequency components. However, if we can transform the high-frequency components in the signal into smooth low-frequency signals convenient for neural network fitting, by some means such as the application of frequency shift technology, we can approximate the original signal via component-by-component fitting. Following this idea, in 2020, Cai et al. proposed the PhaseDNN method to implement a fitting scheme for signals with high-frequency components [14]. As shown in Figure 2, the PhaseDNN method involves four steps: (1) all high-frequency components in the original objective signal are extracted; (2) each high-frequency component is converted into a low-frequency signal using frequency shift technology; (3) different neural networks with identical structure are used to fit these low-frequency signals in parallel; and (4) inverse frequency shift operations are performed for all network predictions to obtain approximated high-frequency components, which are further summed up to recover the original signal.

Figure 3 shows a comparison between the fitted results from the PhaseDNN method and those obtained via vanilla fitting. As shown in Figure 3a, vanilla fitting cannot recover high-frequency components well. If the signal oscillates even faster, the vanilla fitting becomes entirely ineffective. On the contrary, as shown in Figure 3c,d, the recent PhaseDNN method has shown obvious advantages in being able to completely characterize the information of all frequency bands of the signal.

It should be noted that the extracted frequency bandwidth needs to be minimized to ensure fitting accuracy. On the one hand, the smaller the extracted frequency band’s width Δ*ω*, the higher the fitting accuracy. On the other hand, a smaller Δ*ω* indicates more frequency bands are extracted, which directly leads to an increase in computational overhead. When using PhaseDNN for broadband signal fitting, it is often necessary to extract all frequency bands and fit them separately. Therefore, there is a balance between accuracy and computational overhead. For example, consider the task of performing neural network fitting on a signal with a bandwidth of 300 Hz. When Δ*ω* = 10, considering the existence of both real and imaginary parts in the spectrum and the conjugate symmetry, 30 × 2 groups of neural networks need to be trained, which is an acceptable computational overhead. However, for the task of neural network fitting on a signal with a bandwidth of 3000 Hz, 3000 × 2 groups of neural networks need to be trained, which significantly increases the computational overhead.

### 2.2. The Proposed PFF-DNN Method

It is preferable to determine a method that can ensure fitting accuracy without requiring a large number of extracted frequency signals to be selected. Using the fast Fourier transformation (FFT) of broadband signals [15], the digital spectrum of the signal can be obtained much more efficiently. It is conceivable that if we perform a piecewise fitting on a signal’s digital spectrum, no computational overhead is required for frequency selection or frequency shift. This avoids several redundant operations in the PhaseDNN method. In addition, when the frequency spectrum of the signal is not overcomplicated, the efficiency of fitting in the frequency domain will significantly improve. The method proposed here is abbreviated as the PFF-DNN method, and the details of its construction are as follows:

The objective signal is denoted as a real-valued *f*(*x*) in the domain of [*x*_0_, *x_n_*_−1_]. From a digital sampling system, one can obtain the discrete value of the signal *f*(*x*) at the sampling points {*x*_0_, *x*_1_, …, *x_n_*_−1_}. Here, the sampling points are assumed to be uniformly distributed on the interval of [*x*_0_, *x_n_*_−1_], and the discrete value of the signal *f*(*x*) at the sampling points are denoted as *f*_0_, *f*_1_, …, *f_n_*_−1_. One can calculate the frequency spectrum *F*(*ω*) as:(1)$$F\left(\omega_{k}\right)=F_{k}=\sum_{j=0}^{n-1}f_{j}e^{-i2\pi kj/n},$$
where *k* = 0, 1, …, *n* – 1, and *ω_k_* is evenly distributed. The adjacent interval between *ω_k_* depends on the sampling interval. *F*(*ω*) is conjugate symmetric when the signal is real-valued. Here, {*F_k_*} is divided into m segments of length Δ*ω* in order, in which *n* = *m*Δ*ω*:(2)$${\left\{F_{k}\right\}}_{k=0,1,...,n-1}=\overset{m-1}{\underset{i=0}{\cup }}S_{i},$$
where
(3)$$S_{i}={\left\{F_{k}\right\}}_{k=i\Delta \omega ,i\Delta \omega +1,...,\left(i+1\right)\Delta \omega -1},$$

In the following, we define *S_i_* as the *i*th segment of *F*(*ω*). Considering the conjugate symmetry, only half of {*F_k_*} needs fitting. Further considering that the sampling frequency is much larger than the bandwidth of the signal (*ω_b_* = *b*Δ*ω*), Equation (2) can be approximated by:(4)$${\left\{F_{k}\right\}}_{k=0,1,...,n/2}\approx \left(\overset{b}{\underset{i=0}{\cup }}S_{i}\right)\cup \left(\overset{m/2}{\underset{i=b+1}{\cup }}O_{i}\right),$$
where
(5)$$O_{i}=\underset{\Delta \omega }{\underbrace{\left\{0,0,...,0\right\}}},$$
and
(6)$$\begin{array}{cc} F_{n-k}=F_{k}^{*} & k=1,\;2,\;...,\;\left(n-1\right)/2 \end{array},$$
in which * stands for the conjugate operation. For each slice *S_i_*, a neural network is used to approximate the information *T_i_* contained. More precisely:(7)$$T_{i}\left(k\right)\approx \{\begin{array}{ll} F_{k} & k=i\Delta \omega ,\;i\Delta \omega +1,\;...,\;i\Delta \omega +\Delta \omega -1\\ 0 & \mathrm{otherwise} \end{array},$$
where *i* = 1, 2, …, *b*. One can see from Equation (7) that, different from PhaseDNN [14], in the proposed method, each neural network *T_i_* is used to “memorize” the discrete data in the frequency domain. After obtaining these trained neural networks and their predictions {*T_i_*}, the following concatenation operation is used to obtain the approximation of *F*(*ω*):(8)$$F_{k}\approx T\left(k\right)=\{\begin{array}{ll} \sum_{i=0}^{b}T_{i}\left(k\right) & k=0,\;1,\;...,\;\left(n-1\right)/2\\ \sum_{i=0}^{b}T_{i}^{*}\left(n-k\right) & k=\left(n+1\right)/2,\;\left(n+1\right)/2+1,\;...,\;n-1 \end{array},$$

Finally, each sampling signals *f_j_* can be approximated using the following inverse FFT (IFFT):(9)$$f_{j}\approx \frac{1}{n}\sum_{k=0}^{n-1}T\left(k\right)\;e^{i2\pi jk/n},$$

Let us compare the recent PhaseDNN method and the proposed PFF-DNN method in terms of ease of operation. In PhaseDNN, 2*m* convolutions are required to acquire all frequency-selected signals. Then, 2*m* frequency shifts and 2*m* inverse frequency shifts are also involved. In comparison, the proposed method is more straightforward, and only one FFT and one IFFT are required. No frequency selection, frequency shifts, nor inverse shifts are required.

In the PhaseDNN method, DNNs are used to fit continuous signals in the time domain. On the contrary, in the proposed method, DNNs are used to “memorize” discrete data points in the frequency domain. For many commonly used signals (including the signals recommended in [14] to demonstrate the effectiveness of the PhaseDNN method), detailed numerical comparisons are conducted to show that the neural network “memorizes” such discrete frequency values much faster.

## 3. Numerical Experiments

### 3.1. Experimental Setting

Six typical signal analyses were performed to evaluate the fitting performance of our PFF-DNN method, accompanied by a comparison with the existing PhaseDNN method. The signals to be analyzed included some periodic and non-periodic analog signals in the form of explicit functions and complex signals described by the classical Mackey–Glass differential dynamic system. For each signal, detailed comparative evaluations were conducted to describe the effectiveness and efficiency of the different methods for fitting the broadband signals with high-frequency components. The detailed comparisons included (1) convergence curves (the changes of the loss *R* concerning the number of updates *N* of the neural network weights); (2) the convergence process of different frequency bands; (3) the influence of different Δ*ω* on the convergence speed; and (4) the influence of different updating times *N* on the fitting accuracy.

Before showing the experimental results, we first introduce the neural network model and training-related parameters for all subsequent experiments. The neural network structure used in this study was consistent with the network used in [14] (i.e., the 1–40–40–40–40–1 fully connected neural network). Each layer (except the output layer) used the ‘ELU’ activation function [16]. The neural network training adopted the Adam optimizer [17] with a learning rate of 0.0002. The length of each training batch was 100. For the method in [14], 5000 training data and 5000 testing data were evenly distributed across the domain. For the method proposed in this paper, each slice’s length of the training sample depended on the slice’s length in the frequency domain. For both methods, we recorded the approximation error in the root mean square error (RMSE) when Δ*ω* of the slice in the frequency domain was 11, 21, 31, 41, or 51, and when the total number of updates were from 1 to 10,000. The Δ*ω* used in the literature [14] was 11. Therefore, we gradually increased Δ*ω* from this value. When Δ*ω* increases to infinite, both methods will degenerate into the vanilla fitting method. Therefore, we set the upper limit of Δ*ω* to 51. The numerical experiments were completed using a PC with an Intel-core-i9-9900X CPU @ 3.50 GHz with 128 GB-RAM and a Titan X GPU.

### 3.2. Signals with Explicit Expression

#### 3.2.1. Sine on Polynomial Signal

Here, the signal of sine on polynomial was used to test the neural network’s approximation accuracy, which is similar to the function used in the literature [18]. It is described by Equation (10), whose shape and the corresponding frequency spectrum are shown in Figure 4. The signal consists of a high-frequency periodic component and a low-frequency non-periodic component.
(10)$$f\left(x\right)=0.1x^{3}-0.1x^{2}-0.5x+0.3+\sin\left(50x\right),$$

As shown in Figure 5, only the low-frequency component can be approximated if we use the vanilla fitting method. No matter how large *N* is, the high-frequency component cannot be fitted.

On the contrary, if the PhaseDNN or PFF-DNN method was used, the signal was fitted very well. As mentioned earlier, a larger Δ*ω* indicates fewer neural networks used for fitting, which is desirable. For the current example, the bandwidth of the signal is 50 Hz. When Δ*ω* = 11, to approximate the signal precisely, one needs to use at least five networks for fitting. In comparison, when Δ*ω* = 51, one only needs two networks for fitting, which saves 80% of the computation resources. However, an increase in Δ*ω* makes the fitting harder.

Figure 6 shows how the convergence process of the PhaseDNN and PFF-DNN methods changed with Δ*ω*. It was shown that both algorithms converged fast when Δ*ω* was small. However, when Δ*ω* increased, the convergence speed of both methods slowed down. Note that, compared with the PhaseDNN method, the PFF-DNN algorithm was less sensitive to the increase in Δ*ω*. In detail, for the PhaseDNN method, when Δ*ω* = 11, the network updated 10,000 times to reach convergence. However, when Δ*ω* = 41, it was difficult to converge even if it was trained 100,000 times. In contrast, for the PFF-DNN method, when Δ*ω* = 41, it converged within only 2000 updates. That is to say, when Δ*ω* tripled, the convergence time only doubled. This is desirable in practical applications.

Figure 7 shows the fitting results of different frequency bands. Since the frequency spectrum for the neural network to approximate is not overcomplicated, the performance of the PFF-DNN method was better than that of the PhaseDNN method.

Figure 8 shows the influence of Δ*ω* on the algorithm performance. As shown in Figure 8c,f, when Δ*ω* = 11, both methods fit the signal well. However, when Δ*ω* increased to 31, the PhaseDNN method began to show fitting errors. When Δ*ω* = 51, the PhaseDNN method was hardly able to fit the signal effectively. In comparison, the PFF-DNN method was able to still accurately fit the signal.

Figure 9 shows how the number of updates *N* influences the algorithm performance. In the beginning, both algorithms were unable to fit the high-frequency components very well. The comparison between Figure 9b,e shows that the PFF-DNN method converged faster compared with the PhaseDNN method. The red curves in Figure 6 also confirm this point.

#### 3.2.2. ENSO Signal

The signal determined by the following equation is often used to approximate the ENSO data set [19]:(11)$$\begin{array}{ll} f\left(x\right)= & 4.7\cos\left(2\pi x/12\right)+1.1\sin\left(2\pi x/12\right)\\ & +0.2\cos\left(2\pi x/1.7\right)+2.7\sin\left(2\pi x/1.7\right)\\ & +2.1\cos\left(2\pi x/0.7\right)+2.1\sin\left(2\pi x/0.7\right)\\ & -0.5 \end{array}$$

This signal is more complicated than the one described by Equation (10). The shape of the signal and its corresponding frequency spectrum is shown in Figure 10.

According to the trend of the solid red line and the red dashed line in Figure 11, it was shown that the convergence speed of the PFF-DNN method was about an order of magnitude larger. With the increase in Δ*ω*, the convergence time of the PFF-DNN method only doubled. Thus, the larger Δ*ω*, the more training time the PFF-DNN method saves as compared to the PhaseDNN method. Figure 12 and Figure 13 visually show the convergence process of the two networks.

#### 3.2.3. Signal with Increasing Frequency

Next, a signal with fast frequency changes was analyzed. This signal is often used in system identification [20], and its shape and frequency spectrum are shown in Figure 14. The explicit expression of the signal used here is:(12)$$f\left(x\right)=\cos\left[\pi \left(f_{0}+\frac{f_{T}-f_{0}}{T}x^{3}\right)x^{3}\right],$$
in which, *T* = 1, *f*_0_ = 0.01, and *f_T_* = 50. Figure 14a shows that as *x* increases, the oscillation frequency of the signal increases sharply. Figure 14b shows that the amplitude and the oscillation frequency of the signal’s spectrum gradually decrease.

From the curve in Figure 15, it is shown that when Δ*ω* = 11, the efficiency and accuracy of the PFF-DNN method were better than that of the PhaseDNN method. When Δ*ω* increased, the convergence speed of both methods slowed down rapidly. Note that the convergence speed of the PFF-DNN network was relatively faster. Figure 16 and Figure 17 visually compare the convergence process of the two methods.

#### 3.2.4. Piecewise Signal

The piecewise signal was used in [14] to test the neural networks’ performance. The shape and its corresponding frequency spectrum are shown in Figure 18. The signal is described by:(13)$$f\left(x\right)=\{\begin{array}{ll} 10\left(\sin x+\sin3x\right) & x\in \left[-\pi ,0\right]\\ 10\left(\sin23x+\sin137x+\sin203x\right) & x\in \left[0,+\pi \right] \end{array},$$

The spectrum of this signal has several obvious spikes and many slight oscillations. Figure 19 shows that for such a signal, the accuracy of the proposed method was much less sensitive to changes in Δ*ω*. Figure 20 and Figure 21 visually compare the convergence process of the two networks.

#### 3.2.5. Square Wave Signal

The following example was used to test the proposed method’s approximation accuracy on discontinuous signals such as square waves [14]. The shape and its corresponding frequency spectrum are shown in Figure 22. The signal is described by:(14)f(x)=sin(x)+sign[sin(13x)]+sign[sin(23x)]+sign[sin(47x)],

The spectrum of this signal is composed of many irregular spikes. From Figure 23, it is shown that the increase in Δ*ω* had a limited impact on the PFF-DNN method but had a significant impact on the PhaseDNN method. Figure 24 and Figure 25 visually compare the convergence process of the two networks.

### 3.3. Dynamic System

The Mackey-Glass equation is a time-delayed differential equation first proposed to model white blood cell production [21]. This equation has been used to test signal approximations in many related works [22,23,24]. It is described by the solution of the following lagged differential equation:(15)$$\dot{y}\left(x\right)=\frac{0.2y\left(x-30\right)}{1+y^{10}\left(x-30\right)}-0.1y\left(x\right),$$

The shape of the signal and its corresponding frequency spectrum is shown in Figure 26.

As shown, this signal’s frequency spectrum is more complicated. It is distributed irregularly within the interval of [0, 200]. Thus, this problem is relatively hard for both the PhaseDNN and PFF-DNN method.

Figure 27 shows that when Δ*ω* = 21 and Δ*ω* = 41, the training advantage of the PFF-DNN over the PhaseDNN method was relatively obvious. However, when Δ*ω* = 11 and Δ*ω* = 51, the advantage was less obvious. Figure 28 and Figure 29 show that in the middle of training, the convergence speed of the PFF-DNN method was also slightly better.

From this example, it is observed that when the signal’s frequency spectrum is relatively complicated, the performance gain of the PFF-DNN method was not significant. Therefore, we believe that the PFF-DNN and PhaseDNN methods may have their advantages for different applications.

### 3.4. Further Discussions

From the above experiments of fitting a large number of typical signals of different categories, it was observed that the proposed method had two advantages when the frequency domain distribution of the signal is smoother: (1) the proposed method had obvious training efficiency advantages without losing accuracy; (2) the proposed method was less sensitive to the choice of bandwidth. On the surface, this was due to the smoothness of the signal frequency domain distribution, but the deeper reason comes from the difference in complexity of the same segment of the signal in different domains.

In the following, we explain why the PFF-DNN method performed better in the above examples from the perspective of the spectral bias of neural networks. To this end, Figure 30 shows how the PhaseDNN and PFF-DNN methods approximated the third frequency band when Δ*ω* = 11 during the training process of the example in Section 3.2.3. Figure 30a,b shows that the PhaseDNN method essentially handled a continuous signal in the time domain, while the PFF-DNN method essentially interpolated a frequency domain discrete signal. Figure 30d,f shows the approximated results from the PFF-DNN method as a continuous function, which means that the PFF-DNN method interpolated the discrete spectrum with a smooth curve. Compared with Figure 30c,e, it is observed that the curve approximated by the PhaseDNN method oscillated more frequently in the time domain. Thus, it was more difficult for the neural network to learn the signals in the time domain by considering the spectral bias. This discussion may explain why the PFF-DNN method performed better than the PhaseDNN method for these examples.

## 4. Conclusions

Due to the existence of spectral bias, problems occur when a neural network is used for the approximation of broadband signals. That is, the high-frequency parts are hard to fit. For this reason, people often use frequency-selective filters to extract signals in different frequency bands and then use different neural networks to fit them. However, we found in a large number of numerical experiments that this method is inefficient for fitting signals with broad and smooth spectrums. For this reason, this paper proposed a novel alternative method based on parallel fitting in the frequency domain by utilizing frequency domain analysis of broadband signals and the spectral bias nature of neural networks. A substantial improvement in efficiency was observed in extensive numerical experiments. Thus, the PFF-DNN method is expected to become an alternative solution for broadband signal fitting.

In the future, we will further investigate how to determine whether the same signal is easier to fit by a neural network in the time domain or frequency domain and extend our findings to more complex representation spaces.

## Figures and Tables

**Figure 1 sensors-22-07347-f001:**
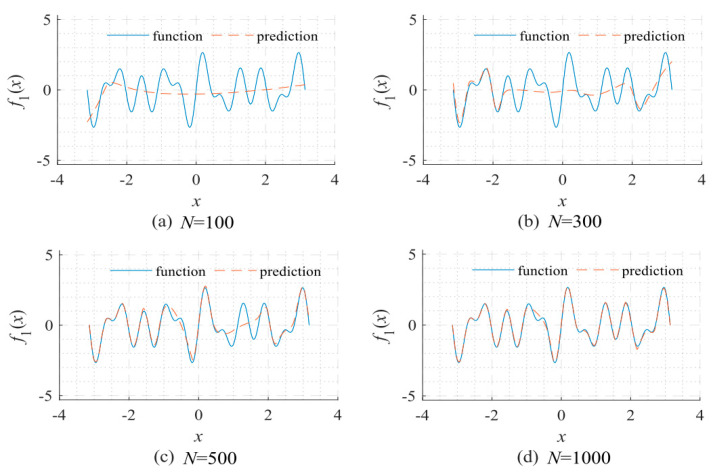
Network fitting results given a different number of updates reflects that neural networks often learn low-frequency components first in the training process: (**a**) *N* = 100; (**b**) *N* = 300; (**c**) *N* = 500; and (**d**) *N* = 1000. The objective function for fitting: *f*_1_ = sin5*x* + sin7*x* + sin11*x*. The network used a 1-40-40-40-40-1 fully connected structure type. The Adam optimizer was used for optimization under the default learning rate.

**Figure 2 sensors-22-07347-f002:**
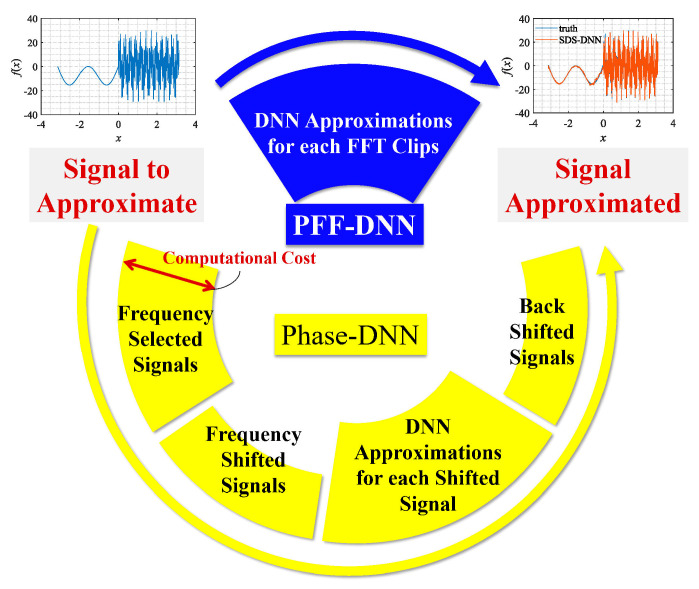
Flowchart of the PhaseDNN algorithm and the proposed PFF-DNN method.

**Figure 3 sensors-22-07347-f003:**
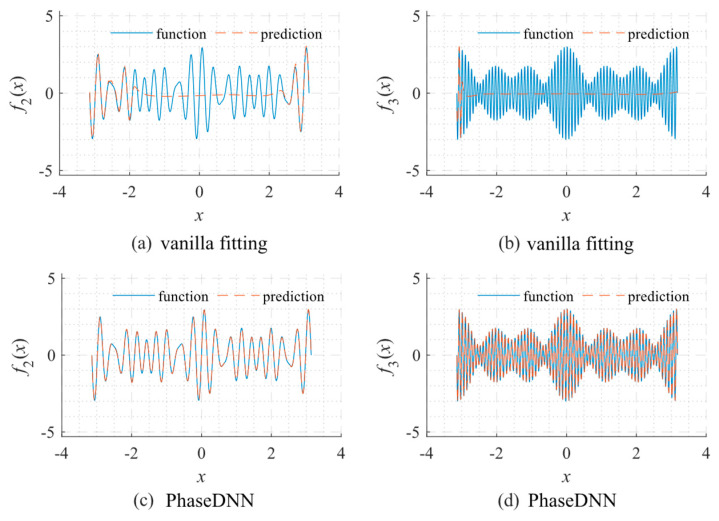
Comparison between the fitted results using the PhaseDNN method and vanilla fitting when *N* = 1000. The objective functions for fitting: *f*_2_ = sin17*x* + sin19*x* + sin23*x* and *f*_3_ = sin67*x* + sin71*x* + sin73*x*. (**a**) Fitting *f*_2_ via vanilla fitting; (**b**) fitting *f*_3_ via vanilla fitting; (**c**) fitting *f*_2_ via PhaseDNN; and (**d**) fitting *f*_3_ via PhaseDNN. The network used a 1-40-40-40-40-1 fully connected structure type. The Adam optimizer was used for optimization under the default learning rate.

**Figure 4 sensors-22-07347-f004:**
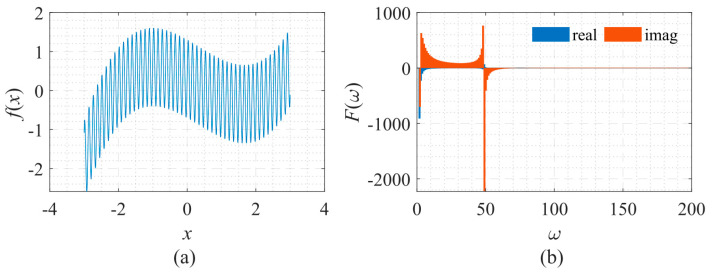
Signal described by Equation (10) with its frequency spectrum. (**a**) The function to be fitted. (**b**) The spectrum of the function.

**Figure 5 sensors-22-07347-f005:**
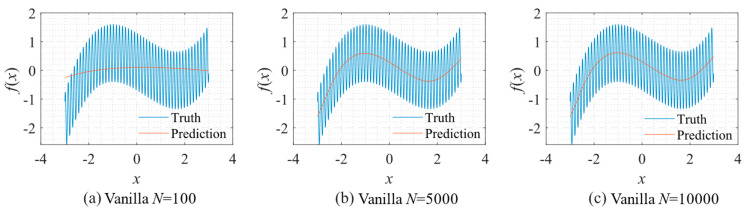
Function fitting using vanilla fitting: (**a**) *N* = 100; (**b**) *N* = 5000; and (**c**) *N* = 10,000.

**Figure 6 sensors-22-07347-f006:**
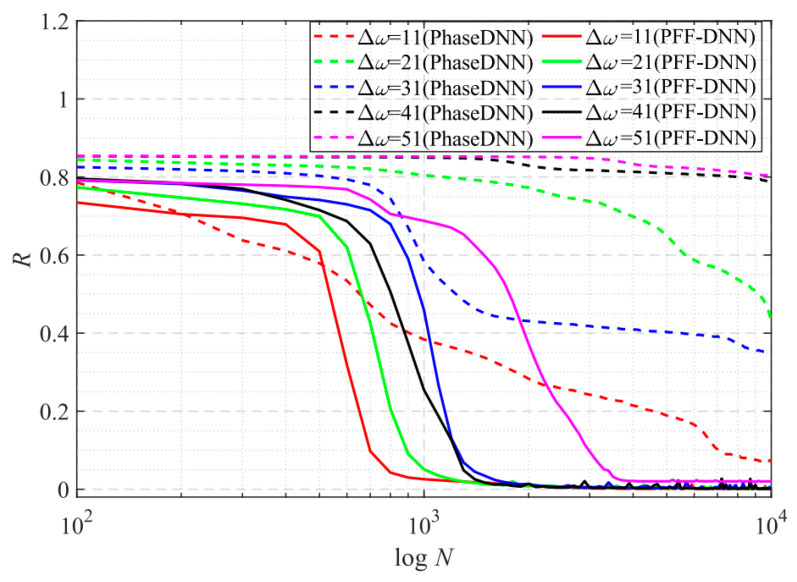
The influence of Δ*ω* on the convergence process.

**Figure 7 sensors-22-07347-f007:**
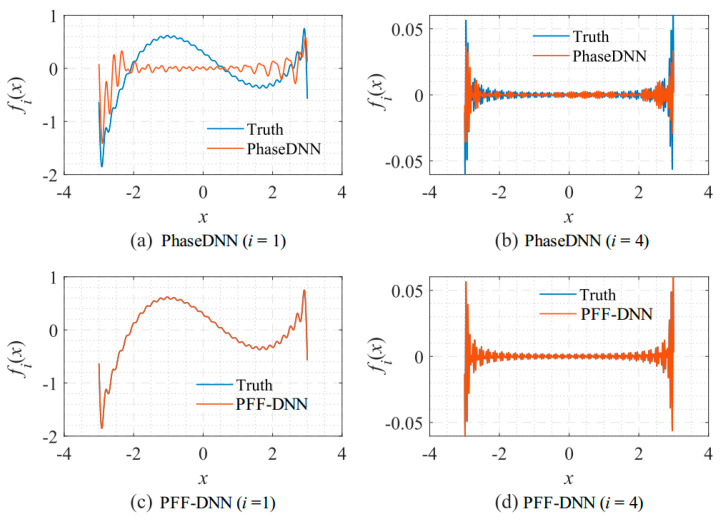
Comparison between the fitting results of the PhaseDNN and PFF-DNN methods for different frequency bands. The fitting results after neural networks were updated 5000 times given Δ*ω* = 31: (**a**) first frequency band (PhaseDNN); (**b**) fourth frequency band (PhaseDNN); (**c**) first frequency band (PFF-DNN); and (**d**) fourth frequency band (PFF-DNN).

**Figure 8 sensors-22-07347-f008:**
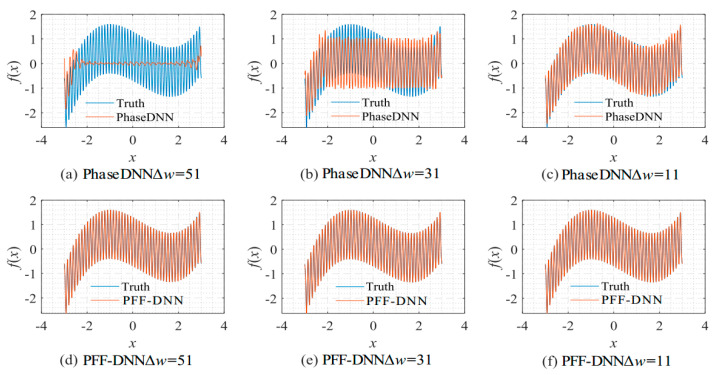
The influence of Δ*ω* on the algorithm performance. The fitting results of the PhaseDNN method after 10,000 updates: (**a**) Δ*ω* = 51; (**b**) Δ*ω* = 31; and (**c**) Δ*ω* = 11. The fitting results of the PFF-DNN method after 10,000 updates: (**d**) Δ*ω* = 51; (**e**) Δ*ω* = 31; and (**f**) Δ*ω* = 11.

**Figure 9 sensors-22-07347-f009:**
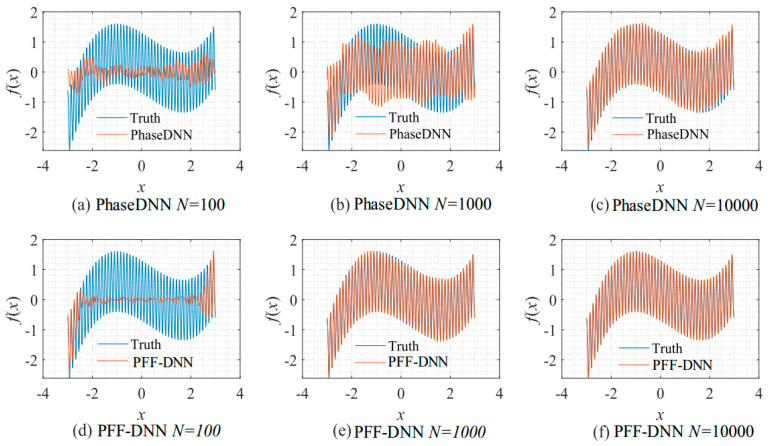
The influence of *N* on the algorithm performance. Fitting results of the PhaseDNN method when Δ*ω* = 11: (**a**) *N* = 100; (**b**) *N* = 1000; and (**c**) *N* = 10,000. Fitting results of the PFF-DNN method when Δ*ω* = 11: (**d**) *N* = 100; (**e**) *N* = 1000; and (**f**) *N* = 10,000.

**Figure 10 sensors-22-07347-f010:**
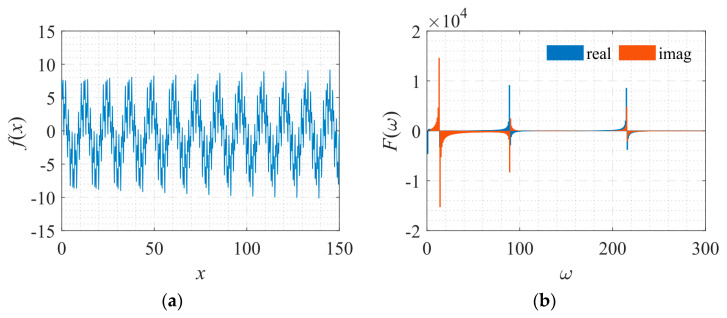
Signal described by Equation (11) with its frequency spectrum. (**a**) The function to be fitted. (**b**) The spectrum of the function.

**Figure 11 sensors-22-07347-f011:**
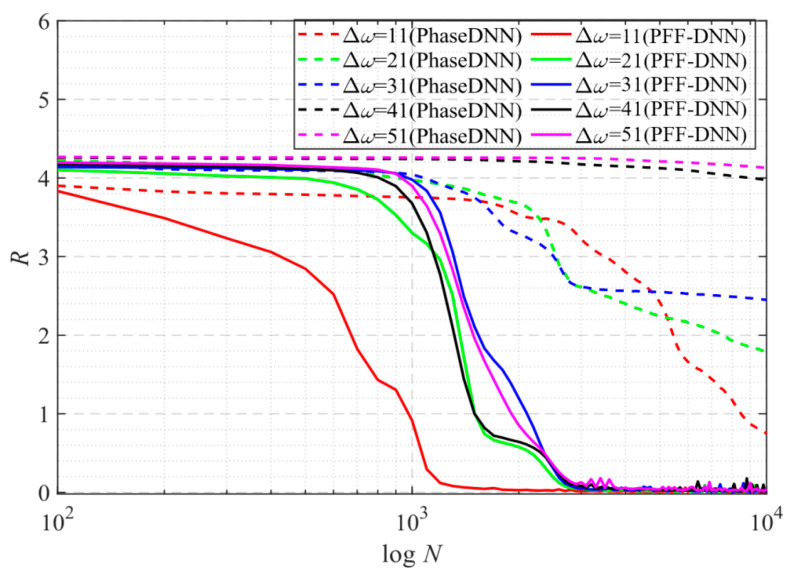
The influence of Δ*ω* on the convergence process.

**Figure 12 sensors-22-07347-f012:**
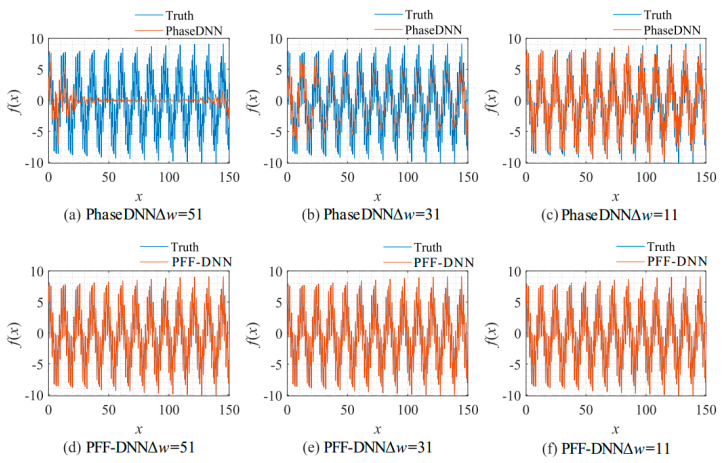
The influence of Δ*ω* on the algorithm performance. The fitting results of the PhaseDNN method after 10,000 updates: (**a**) Δ*ω* = 51; (**b**) Δ*ω* = 31; (**c**) Δ*ω* = 11. The fitting results of the PFF-DNN method after 10,000 updates: (**d**) Δ*ω* = 51; (**e**) Δ*ω* = 31; (**f**) Δ*ω* = 11.

**Figure 13 sensors-22-07347-f013:**
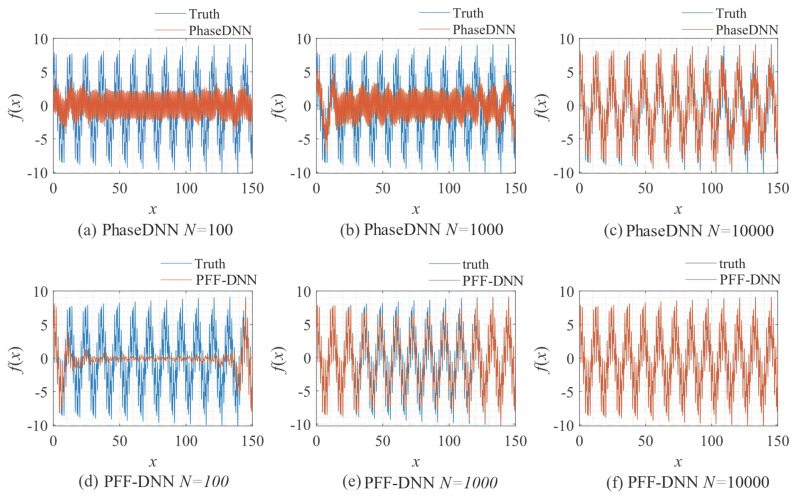
The influence of *N* on the algorithm performance. Fitting results of the PhaseDNN method when Δ*ω* = 11: (**a**) *N* = 100; (**b**) *N* = 1000; and (**c**) *N* = 10,000. Fitting results of the PFF-DNN method when Δ*ω* C= 11: (**d**) *N* = 100; (**e**) *N* = 1000; and (**f**) *N* = 10,000.

**Figure 14 sensors-22-07347-f014:**
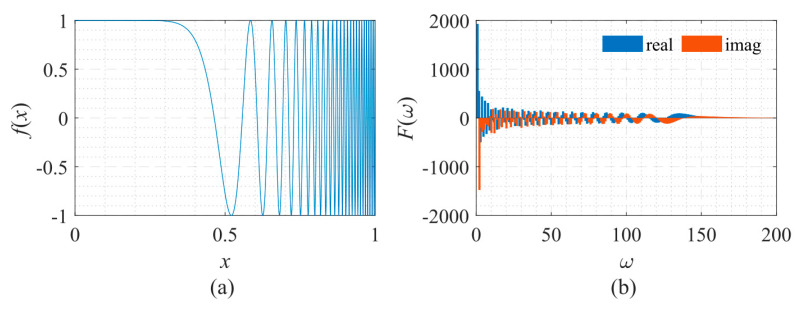
Signal described by Equation (12) with its frequency spectrum. (**a**) The function to be fitted. (**b**) The spectrum of the function.

**Figure 15 sensors-22-07347-f015:**
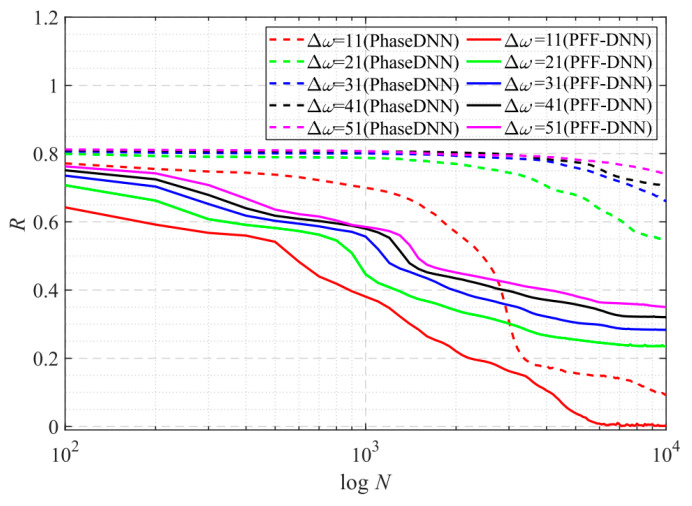
The influence of Δ*ω* on the convergence process.

**Figure 16 sensors-22-07347-f016:**
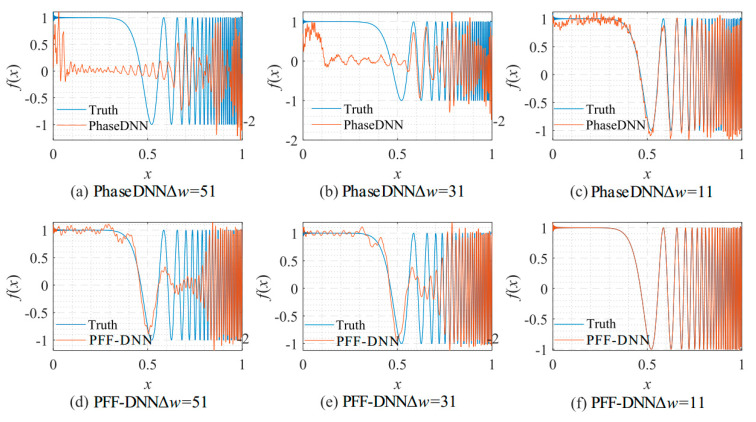
The influence of Δ*ω* on the algorithm performance. The fitting results of the PhaseDNN method after 10,000 updates: (**a**) Δ*ω* = 51; (**b**) Δ*ω* = 31; and (**c**) Δ*ω* = 11. The fitting results of the PFF-DNN method after 10,000 updates: (**d**) Δ*ω* = 51; (**e**) Δ*ω* = 31; and (**f**) Δ*ω* = 11.

**Figure 17 sensors-22-07347-f017:**
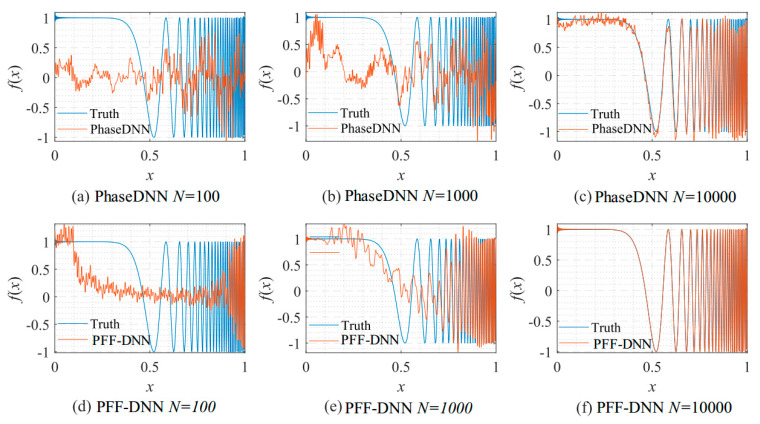
The influence of *N* on the algorithm performance. Fitting results of the PhaseDNN method when Δ*ω* = 11: (**a**) *N* = 100; (**b**) *N* = 1000; and (**c**) *N* = 10,000. Fitting results of the PFF-DNN method when Δ*ω* = 11: (**d**) *N* = 100; (**e**) *N* = 1000; and (**f**) *N* = 10,000.

**Figure 18 sensors-22-07347-f018:**
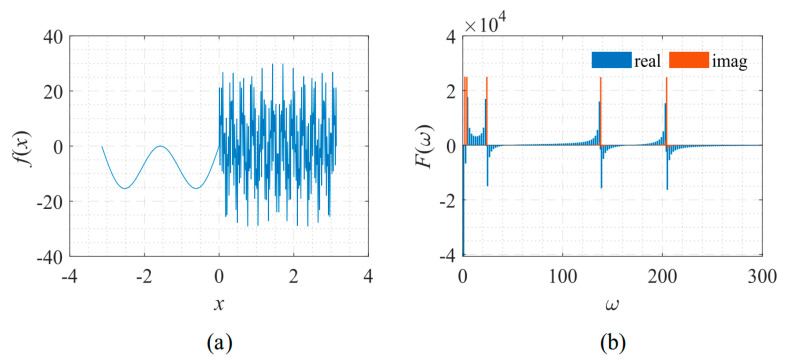
Signal described by Equation (13) with its frequency spectrum. (**a**) The function to be fitted. (**b**) The spectrum of the function.

**Figure 19 sensors-22-07347-f019:**
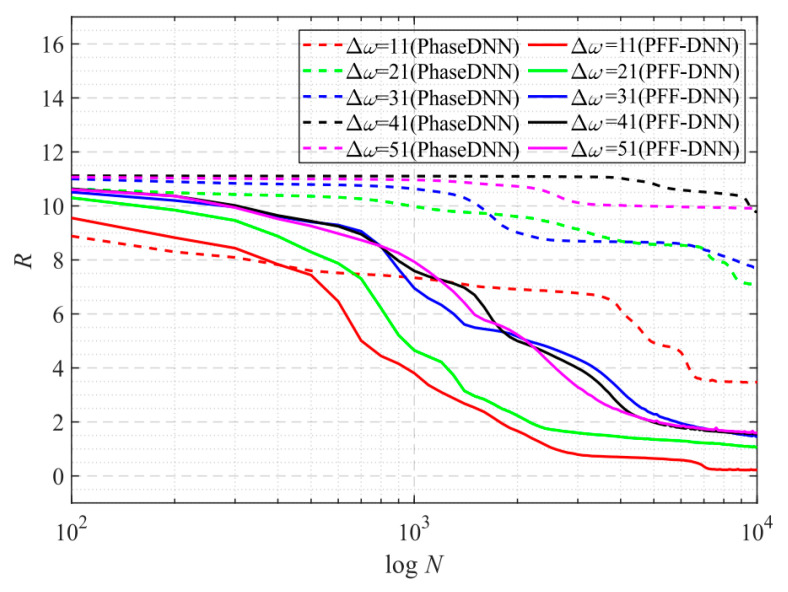
The influence of Δ*ω* on the convergence process.

**Figure 20 sensors-22-07347-f020:**
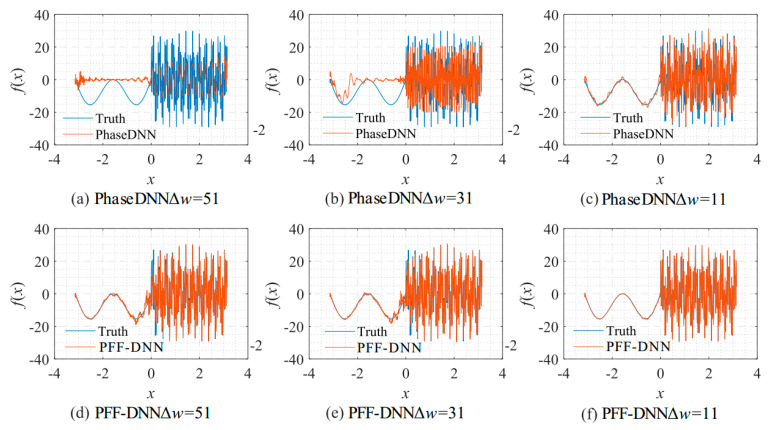
The influence of Δ*ω* on the algorithm performance. The fitting results of the PhaseDNN method after 10,000 updates: (**a**) Δ*ω* = 51; (**b**) Δ*ω* = 31; and (**c**) Δ*ω* = 11. The fitting results of the PFF-DNN method after 10,000 updates: (**d**) Δ*ω* = 51; (**e**) Δ*ω* = 31; and (**f**) Δ*ω* = 11.

**Figure 21 sensors-22-07347-f021:**
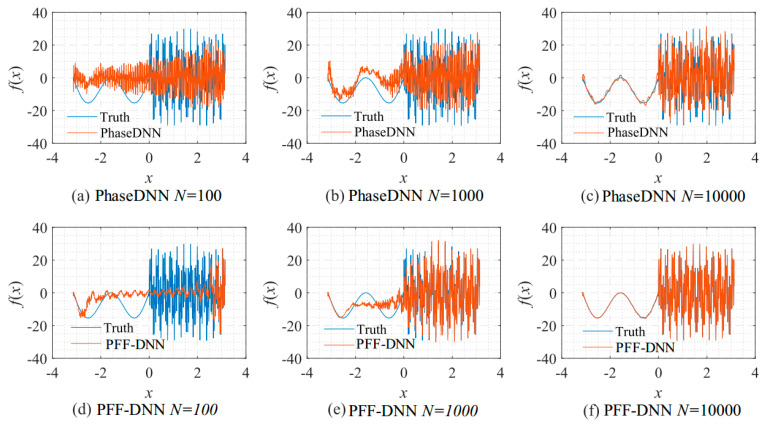
The influence of *N* on the algorithm performance. Fitting results of the PhaseDNN method when Δ*ω* = 11: (**a**) *N* = 100; (**b**) *N* = 1000; and (**c**) *N* = 10,000. Fitting results of the PFF-DNN method when Δ*ω* = 11: (**d**) *N* = 100; (**e**) *N* = 1000; and (**f**) *N* = 10,000.

**Figure 22 sensors-22-07347-f022:**
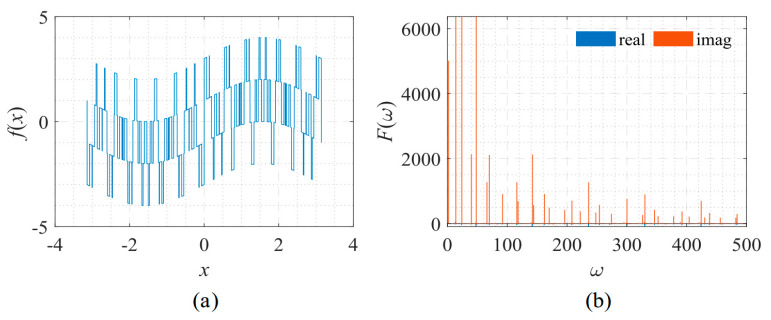
Signal described by Equation (14) with its frequency spectrum. (**a**) The function to be fitted. (**b**) The spectrum of the function.

**Figure 23 sensors-22-07347-f023:**
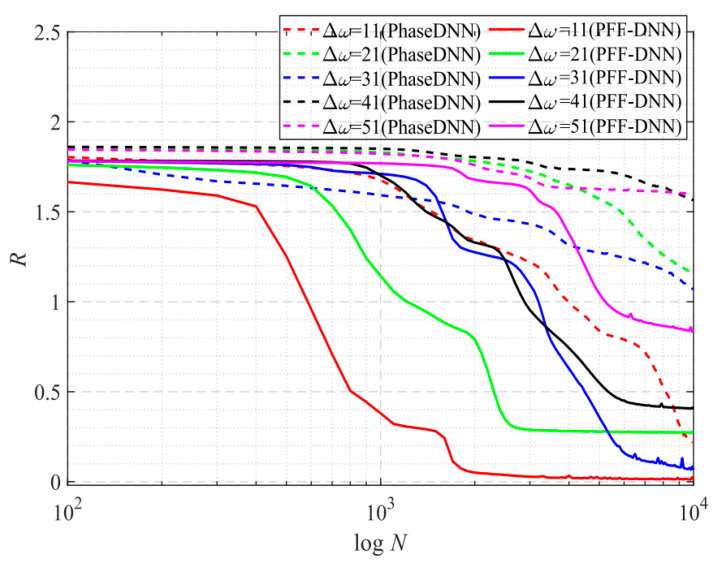
The influence of Δ*ω* on the convergence process.

**Figure 24 sensors-22-07347-f024:**
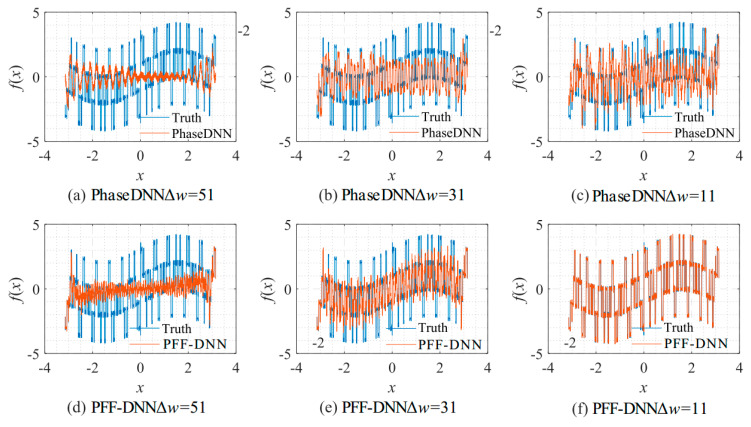
The influence of Δ*ω* on the algorithm performance. The fitting results of the PhaseDNN method after 10,000 updates: (**a**) Δ*ω* = 51; (**b**) Δ*ω* = 31; and (**c**) Δ*ω* = 11. The fitting results of the PFF-DNN method after 10,000 updates: (**d**) Δ*ω* = 51; (**e**) Δ*ω* = 31; and (**f**) Δ*ω* = 11.

**Figure 25 sensors-22-07347-f025:**
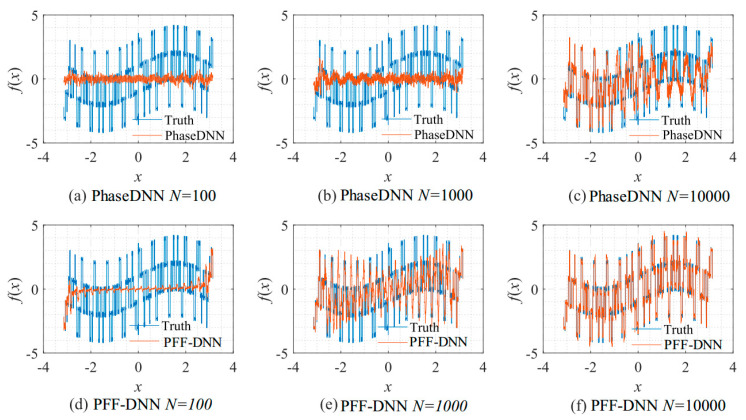
The influence of *N* on the algorithm performance. Fitting results of the PhaseDNN method when Δ*ω* = 11: (**a**) *N* = 100; (**b**) *N* = 1000; and (**c**) *N* = 10,000. Fitting results of the PFF-DNN method when Δ*ω* = 11: (**d**) *N* = 100; (**e**) *N* = 1000; and (**f**) *N* = 10,000.

**Figure 26 sensors-22-07347-f026:**
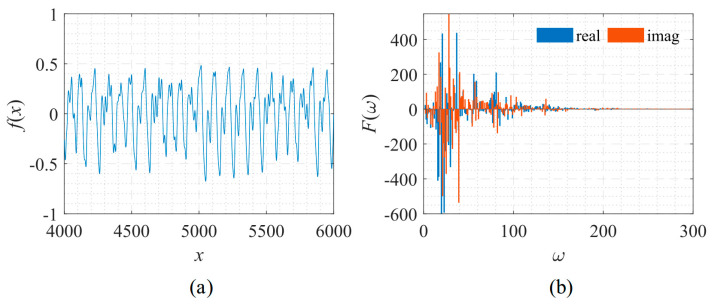
Signal described by Equation (15) with its frequency spectrum. (**a**) The function to be fitted. (**b**) The spectrum of the function.

**Figure 27 sensors-22-07347-f027:**
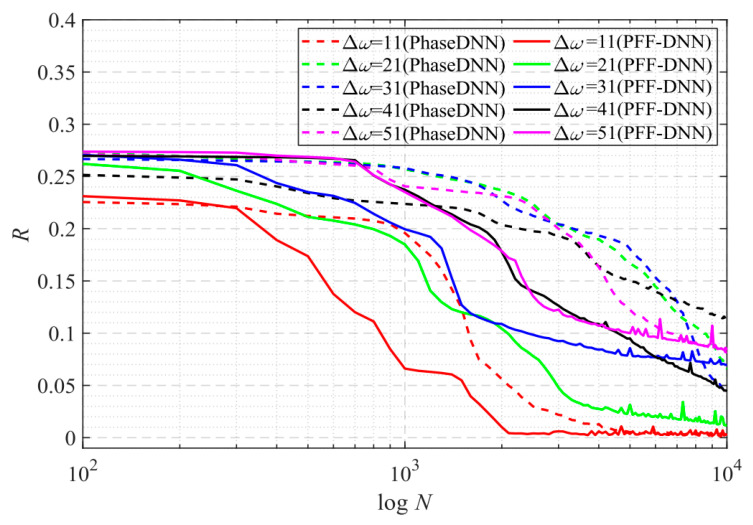
The influence of Δ*ω* on the convergence process.

**Figure 28 sensors-22-07347-f028:**
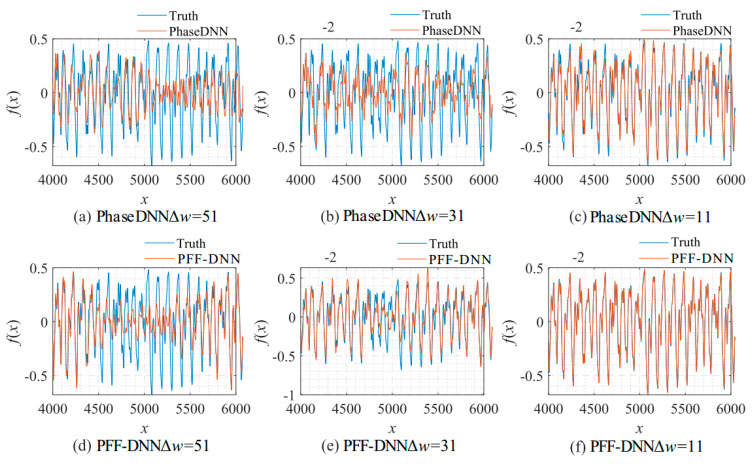
The influence of Δ*ω* on the algorithm performance. The fitting results of the PhaseDNN method after 10,000 updates: (**a**) Δ*ω* = 51; (**b**) Δ*ω* = 31; and (**c**) Δ*ω* = 11. The fitting results the of PFF-DNN method after 10,000 updates: (**d**) Δ*ω* = 51; (**e**) Δ*ω* = 31; and (**f**) Δ*ω* = 11.

**Figure 29 sensors-22-07347-f029:**
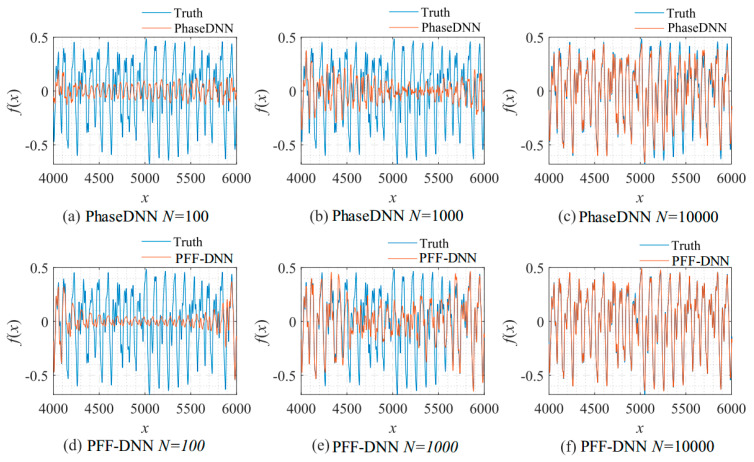
The influence of *N* on the algorithm performance. Fitting results of the PhaseDNN method when Δ*ω* = 11: (**a**) *N* = 100; (**b**) *N* = 1000; and (**c**) *N* = 10,000. Fitting results of the PFF-DNN method when Δ*ω* = 11: (**d**) *N* = 100; (**e**) *N* = 1000; and (**f**) *N* = 10,000.

**Figure 30 sensors-22-07347-f030:**
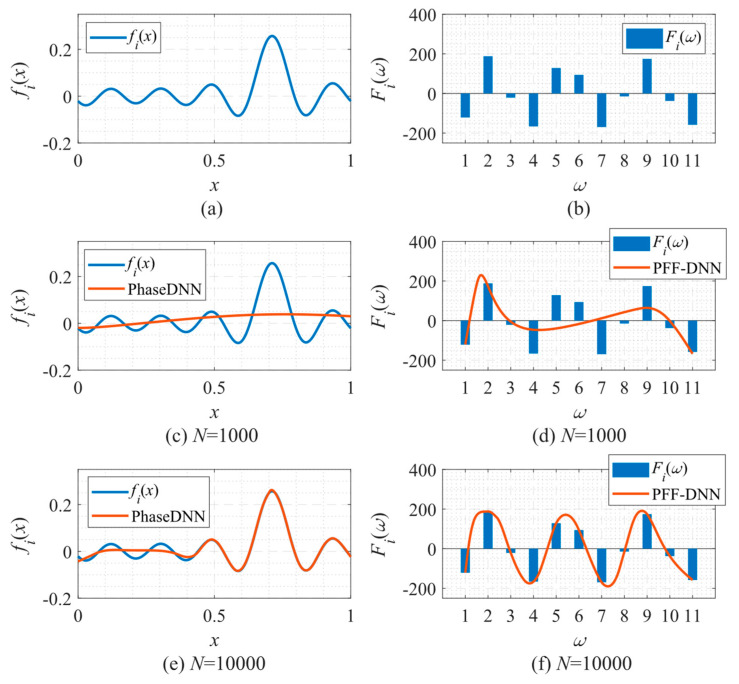
Approximation performance during training. (**a**) The third frequency band of the fitted function (the real part after frequency shift); (**b**) the spectrum (real part) of the function in (**a**); (**c**) fitting results of PhaseDNN after 1000 updates; (**d**) fitting results of PFF-DNN after 1000 updates; (**e**) fitting results of PhaseDNN after 10,000 updates; and (**f**) fitting results of PFF-DNN after 10,000 updates.

## Data Availability

The details of the approach discussed herein and the specific values of the parameters considered have been provided in the paper. Hence, we are confident that the results can be reproduced. Readers interested in the code are encouraged to contact the authors by e-mail.

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
