# Peer review of "Parallel Frequency Function-Deep Neural Network for Efficient Approximation of Complex Broadband Signals"

_sensors, 2022, doi:10.3390/s22197347_

Round 1

Reviewer 1 Report

The work presents a Parallel frequency function-deep neural network for efficient approximation of complex broadband signals. The general concept is good but the writing is quite difficult to follow due to several complex sentences. In some instances, the intended meaning is not well communicated. The authors are required to revise the English language structure and avoid the use of unrelated words. The following comments are provided to improve the paper.

1. The main contribution should be highlighted in bullet form. The paper organization should be added at the end of section 1.

2. The literature review is destitute and a rigorous review of the related state-of-the-art is requested. The gap in the existing works should be outlined and show how the current study has addressed or filled the gaps.

3. The experiment setting is not deployed. The authors should consider a possible test of the projected scheme.

4. The results are not compact. Several graphs were seen littering the paper. Only the most important results should be presented for brevity.

5. The implications of the results should be clearly highlighted and elaborated.

6. A comparison of the projected results is required. The results should be compared to related work to demonstrate the efficacy of the new scheme clearly.

7. The future scope is missing in the conclusion. There is a need to include the future scope of the paper.

8. The reference list is impoverished. The authors support their claims with the most recent papers related to the study.

Author Response

Dear Reviewer,

We have revised our previous manuscript (Manuscript ID: sensors-1908564, titled ‘Parallel frequency function-deep neural network for efficient approximation of complex broadband signals’) according to the your comments and resubmitted it for your consideration of potential publication in Sensor. A detailed description of each comment and how the manuscript was revised are provided below. We thank for your time and effort in carefully reviewing our paper.

Best regards,
Zhi Zeng, Pengpeng Shi, Fulei Ma, Peihan Qi

Reviewer 2 Report

Title: Parallel frequency function-deep neural network for efficient approximation of complex broadband signals

 1. The authors have not defined problem and study area, instead of it they have started discussing about the limitations of previous methodologies. What kind of problem is observed in approximation of complex broadband signals. In addition to this, avoid citing references in the abstract.

2. Must highlight the numerical findings of the proposed study with novelty in the abstract.

3. The authors must justify how they model i.e., parallel frequency function-deep neural network (PFF-DNN) is better than other models with comparative analysis of previous studies.

4. The motivation, research gap, contribution and organization of the study must be included in the introduction section. Appropriate reference must be cited for the figure 1, as it discusses the results of the previous study.

5. The conclusion of the study must highlight with contributions, numerical findings, novelty and originality of the study.

6. The abbreviation is placed after defining its full form at the first instance, however authors have not followed this in several parts of the manuscript (For example in the abstract: PINN & phaseDNN).

Author Response

(The authors gave the same response as above.)

Round 2

Reviewer 1 Report

The authors have addressed most of my earlier comments.

Reviewer 2 Report

authors addressed most of the comments.